# Why Should I Trust You, Bellman? Evaluating the Bellman Objective with Off-Policy Data

## Abstract

In this work, we analyze the effectiveness of the Bellman equation as a proxy objective for value prediction accuracy in off-policy evaluation. While the Bellman equation is uniquely solved by the true value function over all state-action pairs, we show that in the finite data regime, the Bellman equation can be satisfied exactly by infinitely many suboptimal solutions. This means that Bellman error can be minimized without necessarily improving the accuracy of the value function. We find this observation extends to practical settings; when computed over an off-policy dataset, the Bellman error bears little relationship to value prediction error. Consequently, we show that the Bellman error is a poor metric for comparing value functions, and therefore, an ineffective objective for off-policy evaluation. Finally, we discuss differences between Bellman error and the non-stationary objective used by iterative methods and deep reinforcement learning, and highlight how the effectiveness of this objective relies on generalization during training.

## 1 Introduction

In reinforcement learning (RL), value functions are a measure of performance of a target policy. Value functions are an important quantity in RL as they can be used to inform decision-making. Consequently, many modern reinforcement learning algorithms rely on a value function in some capacity (Gu et al., 2016; Schulman et al., 2017; Fujimoto et al., 2018; Badia et al., 2020).

The Bellman equation is a fundamental relationship in RL which relates the value of a state-action pair to the state-action pair that follows, and is uniquely satisfied over all state-action pairs by the true value function. The existence of the Bellman equation suggests a straightforward approach for approximate value function learning, where a function is trained to minimize the Bellman error (the difference of both sides of the equation). The Bellman equation has played a prominent role in many historically significant approaches (Schweitzer & Seidmann, 1985; Baird, 1995; Bradtke & Barto, 1996; Antos et al., 2008; Sutton et al., 2009), as well as the more modern family of deep RL algorithms (Mnih et al., 2015; Lillicrap et al., 2015; Gu et al., 2016; Hessel et al., 2017).

In this work, we examine the relationship between the Bellman equation and the accuracy of value functions. We do so through off-policy evaluation (OPE), which presents the task of learning the value function of a target policy with data gathered from a separate and possibly unknown behavior policy. OPE, which is a subcomponent of virtually any off-policy RL algorithm, is an ideal setting for evaluating value functions as it provides a clear metric of performance (value prediction error) and provides consistency across trials (fixed dataset and target policy).

Our main thesis is that since the Bellman equation is meant to consider the entire MDP and all possible state-action pairs, when it is instead estimated over a finite dataset, there is likely to be some breakdown in its relationship to value prediction. This work aims to better understand that breakdown through theoretical analysis and empirical study. Our key discoveries are under off-policy, function approximations, and finite data assumptions:

**Bellman error is a poor metric for value error.** We find that given two arbitrary value functions, comparing their Bellman error is insufficient to determine which value function is more accurate. This problem is highlighted by experiments which show that value functions trained to minimize

Bellman error directly (Baird, 1995) have lower Bellman error but higher value error, than value functions trained by iterative methods (Ernst et al., 2005). We find that this non-correspondence in relative ordering over error terms holds even when evaluated over on-policy data (Figure 2), and only worsens further with off-policy datasets (Figure 3).

**Bellman error is a poor objective for learning off-policy.** A natural consequence of the Bellman error being a weak metric for value error, is that the Bellman error makes for a poor off-policy objective. Our experiments show that value functions trained by different algorithms exhibit different behaviors. As such, Bellman error cannot be used as a metric for arbitrary value functions. However, we find that when comparing value functions trained by the same algorithm, Bellman error can be used as an accurate measure for value error, but *only if the error terms are evaluated with on-policy data* (Table 1). This means that Bellman error is only a meaningful objective when used on-policy.

**Iterative methods rely on generalization for successful training.** Iterative methods, such as many deep RL algorithms (Mnih et al., 2015; Lillicrap et al., 2015), use a slightly different objective than Bellman error, where the target is assumed to be fixed. This means the objective is non-stationary and evolves during learning. Similar to the Bellman error objective, we find that examining the FQE loss alone is insufficient to determine the accuracy of the value function. However, we remark that we can compare two functions if we take the loss with respect to the same fixed target. This means that if the frozen target is accurate, then the distance to that target is a good proxy for value error. This exposes the reliance of iterative methods to generalization which occurs *during training*.

Our work highlights problems with using Bellman error as a signal, or objective, in the off-policy setting, and aims to provide practitioners a better understanding of Bellman equation-based loss functions, the role of generalization in RL, and the learning dynamics of value functions. Our findings point to an underappreciation of the importance of finite data in widely used objectives and we encourage the community to place a higher emphasis on practical settings.

## 2 BACKGROUND

Reinforcement learning (RL) is an optimization framework for tasks of sequential nature (Sutton & Barto, 1998). Typically, tasks are defined as a Markov decision process $(\mathcal{S}, \mathcal{A}, \mathcal{R}, p, d_0, \gamma)$, with state space $\mathcal{S}$, action space $\mathcal{A}$, reward function $\mathcal{R}$, transition dynamics $p$, initial state distribution $d_0$, and discount factor $\gamma \in [0, 1)$. Actions are selected according to a policy $\pi$.

The performance of a policy is measured by its discounted return $\mathbb{E}_\pi[\sum_t^\infty \gamma^t r(s_t, a_t)]$. Off-policy evaluation (OPE) is the task of approximating the value function $Q^\pi(s, a) = \mathbb{E}_\pi[\sum_t^\infty \gamma^t r(s_t, a_t)|s_0 = s, a_0 = a]$ of a target policy, given samples from an arbitrary dataset. A fundamental relationship regarding value functions is the Bellman equation (Bellman, 1957):

$$Q^\pi(s, a) = \mathbb{E}_{r, s' \sim p, a' \sim \pi} \left[ r + \gamma Q^\pi(s', a') \right], \tag{1}$$

which relates the value of the current state-action pair to an expectation over the next state-action pair. Given an approximate value function $Q$ (distinguished from the true value function $Q^\pi$ by dropping the $\pi$ superscript) of a target policy $\pi$, we denote the Bellman error $\epsilon(s, a)$:

$$\epsilon(s, a) := Q(s, a) - \mathbb{E}_{r, s' \sim p, a' \sim \pi} \left[ r + \gamma Q(s', a') \right]. \tag{2}$$

In policy evaluation, the main objective of interest is value error of a state-action pair $\Delta(s, a)$:

$$\Delta(s, a) := Q(s, a) - Q^\pi(s, a), \tag{3}$$

where $Q^\pi$, the true value function, is intractable without access to the underlying MDP. A standard result is if the Bellman equation converges to the fixed point then the value function must be the true value function. We can re-frame this result in terms of Bellman errors and value errors.

**Proposition 1** *If the Bellman error $\epsilon(s, a) = 0$ for all state-action pairs $(s, a) \in \mathcal{S} \times \mathcal{A}$, then the value error $\Delta(s, a) = 0$ for all state-action pairs $(s, a) \in \mathcal{S} \times \mathcal{A}$.*

In instances where we cannot compute the Bellman error exactly, such as from samples in a non-deterministic environment, we can instead use temporal difference (TD) learning, where the TD error $\delta(i)$ is a sample-based approximation to Bellman error which can be computed over a transition $i := (s, a, r, s')$, $\delta(i) := Q(s, a) - (r + \gamma Q(s', a'))$, where $a'$ is sampled from the policy $\pi$.

Note that the expected TD error is simply the Bellman error $\epsilon(s, a) = \mathbb{E}_{r,s',a'}[\delta(i)]$, where the two values are identical if the environment and policy are deterministic.

In this work we focus on two algorithms based on the Bellman equation, which will update an approximate value function $Q$, using samples from a finite dataset $\mathcal{D}$. Bellman residual minimization (BRM) (Baird, 1995) directly minimizes the Bellman error over samples from the dataset $\mathcal{D}$:

$$\mathcal{L}_{\text{BRM}}(Q) := \frac{1}{|\mathcal{D}|} \sum_{(s,a,r,s') \sim \mathcal{D}, a' \sim \pi} \left(Q(s, a) - (r + \gamma Q(s', a'))\right)^2. \tag{4}$$

Fitted Q-Evaluation (FQE) (Ernst et al., 2005; Le et al., 2019) is an iterative method for minimizing Bellman error:

$$\mathcal{L}_{\text{FQE}}(Q) := \frac{1}{|\mathcal{D}|} \sum_{(s,a,r,s') \sim \mathcal{D}, a' \sim \pi} \left(Q(s, a) - (r + \gamma \bar{Q}(s', a'))\right)^2. \tag{5}$$

The key distinction between the two algorithms is that BRM directly updates both $Q(s, a)$ and $Q(s', a')$, while FQE only considers $Q(s, a)$. This is because FQE uses $\bar{Q}(s', a')$, a target value function which is updated $\bar{Q} \leftarrow Q$ after a fixed number of time steps (possibly including every time step), meaning that only the left side of the Bellman equation is directly updated.

## 3 EXPERIMENTAL DESIGN

Our goal is to thoroughly evaluate the relationship between Bellman error (a measurable proxy) and value error (an unmeasurable true objective) in the case of off-policy evaluation with finite samples. In this section we outline the experimental choices used in our empirical evaluation. Comprehensive experimental details (i.e. hyperparameters, architecture, etc.) can be found in the Appendix E.

**Setting.** We consider the setting of off-policy evaluation (OPE), as it allows to directly compare value functions over a clear metric, value error. Our experiments consider a variety of continuous-action tasks through the MuJoCo simulator (Todorov et al., 2012; Brockman et al., 2016), as it is deterministic and high-dimensional. Determinism in the dynamics is desirable as it, alongside a deterministic policy, makes the Bellman error and TD error identical. This allows us to compute the Bellman error exactly and ignore the double sampling issue for residual gradient methods (Baird, 1995). Value functions are trained to evaluate an expert deterministic target policy from a fully trained TD3 agent (Fujimoto et al., 2018), using a standard discount factor $\gamma = 0.99$.

**Algorithms.** Our experiments are based on Bellman residual minimization (BRM) (Baird, 1995) and Fitted Q-Evaluation (FQE) (Ernst et al., 2005; Le et al., 2019). We use these algorithms due to their popularity in the literature, and to highlight differences in methods which minimize Bellman error directly or indirectly. Network architecture and hyperparameters are the same between algorithms and are selected to match state-of-the-art deep RL methods (Fujimoto et al., 2018; Haarnoja et al., 2018a) for the MuJoCo domain. FQE is implemented using a target network updated with Polyak averaging. In every experiment, algorithms are trained for 1 million time steps and 10 seeds.

**Training Datasets.** Each dataset is collected by using noisy versions of the target policy. This allows us to rank the distribution shift of each dataset. Each noise level corresponds to both the probability of selecting a uniformly random action, as well as the standard deviation of Gaussian noise added to the actions (noting that actions are in the range $[-1, 1]$). We use uniformly random actions to ensure that not all actions are centered around the target policy, and Gaussian noise to ensure that every action is distinct from actions selected by the target policy.

**Metrics.** We use the mean squared Bellman error, as it is the most common objective on the Bellman error (Baird, 1995; Sutton & Barto, 1998). For better interpretability, we use the absolute value error, normalized by dividing by a constant term equal to the average true value function $Q^\pi$ sampled on-policy. As an example, this means that $0.1$ value error roughly corresponds to a percent difference of 10%. Some experiments are repeated in Appendix D with variations of these metrics.

## 4 THE BELLMAN EQUATION AS AN OBJECTIVE

In this section we discuss the role of the Bellman error as a proxy objective for value error. Our main result is that missing transitions break the fundamental relationship between Bellman error

and value error, meaning that one of these error terms can be minimized independently of the other. Consequently, this means that minimizing the empirical Bellman error makes for an ineffective objective, as it does not guarantee a corresponding reduction in value error. We show this problem theoretically and through simple examples (4.1), then demonstrate this phenomenon occurs in standard, widely used benchmark environments (4.2). Finally, we discuss the effectiveness of deep RL methods in spite of these concerns, and highlight the role of generalization in off-policy RL (4.3).

## 4.1 THEORETICAL ANALYSIS

Recall the key idea behind the Bellman equation is that it is uniquely satisfied by the true value function over all state-action pairs. Therefore, if we are interested in off-policy evaluation, the Bellman error is used as a measurable proxy objective to value error, which is typically unmeasurable. While completely minimizing the Bellman error results in the optimal solution, the Bellman error is only a proxy to value error, and does not share an exact correspondence, even when considering the entire MDP. Consider the following proposition.

**Proposition 2** *For any constant $C > 0$ and discount factor $\gamma \in (0,1)$, there exists an MDP and a pair of value functions $(Q_1, Q_2)$ with Bellman errors $(\epsilon_1, \epsilon_2)$ and value errors $(\Delta_1, \Delta_2)$, such that for all state-action pairs $(s,a) \in \mathcal{S} \times \mathcal{A}$, the absolute Bellman error of $Q_2$ is greater than the absolute Bellman error of $Q_1$ by $C$, $|\epsilon_2(s,a)| - |\epsilon_1(s,a)| > C$, but the absolute value error of $Q_1$ is greater than the absolute value error of $Q_2$ by $C$, $|\Delta_1(s,a)| - |\Delta_2(s,a)| > C$.*

This means that a reduction in Bellman error, even over all state-action pairs, does not guarantee a corresponding reduction in value error. To understand how this outcome is possible, consider an infinitely long chain MDP, with reward $r = 0$ for all transitions. Let $k > 0$. Let $Q_1(\cdot) = k/(1 - \gamma)$ for all inputs, and let $Q_2(s_t) = (-1)^t k$, in other words, $k$ on even states and $-k$ on odd states. While $Q_1$ clearly has higher absolute value error, when we look at the Bellman error, the distance between $Q_1(s_t, \cdot)$ and $\gamma Q_1(s_{t+1}, \cdot)$ will be less than $Q_2(s_t, \cdot)$ and $\gamma Q_2(s_{t+1}, \cdot)$ as $Q_2$ swaps signs at each timestep. See the Appendix B.2 for the full details.

This non-correspondence between value error and Bellman error was possible as the absolute value of the Bellman error does not capture the bias in the value error. We can better understand bias in the value error by simply summing the Bellman errors over relevant transitions.

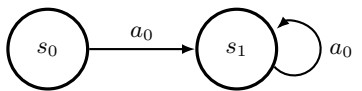

Figure 1: A basic MDP. If $(s_0, a_0)$ is contained in the dataset but $(s_1, a_1)$ is not, by carefully selecting the values $Q(s_0, a_0)$ and $Q(s_1, a_1)$, we can construct examples where the Bellman error of the dataset is 0 but the value error is arbitrarily large.

**Theorem 1** *Let $d^\pi(s', a'|s, a) = (1 - \gamma)\sum_{t=0}^{\infty} \gamma^t p^\pi((s,a) \to s', t)\pi(a'|s')$, be the conditional discounted state-action occupancy, where $p^\pi((s,a) \to s, t)$ is the probability of leaving the state-action pair $(s, a)$ and visiting the state $s$ after $t$ time steps. The value error $\Delta(s, a)$ of a state-action pair $(s, a)$ can be defined as a function of the Bellman error $\epsilon(s', a')$ over $d^\pi(s', a'|s, a)$:*

$$\Delta(s,a) = \frac{1}{1-\gamma}\mathbb{E}_{(s',a')\sim d^\pi(\cdot|s,a)}[\epsilon(s',a')]. \tag{6}$$

A direct consequence of Theorem 1 is the aforementioned uniqueness property of the Bellman equation. That is, if the Bellman error is 0 for all relevant state-action pairs, which may be visited by the target policy, then the value error must also be 0. However, if we are instead examining a finite dataset, this relationship also exposes the concern that if any relevant transitions are missing, then the desired property of a unique solution of the Bellman equation is broken.

**Corollary 1** *If there exists a state-action pair $(s', a')$ not contained in the dataset $\mathcal{D}$, where the state-action occupancy $d^\pi(s', a'|s, a) > 0$, then for any $C > 0$, there exists a value function such that the Bellman error is 0 for all state-action pairs in the dataset $\mathcal{D}$, while the value error of the state-action pair $(s, a)$ is $C$.*

Consider the simple two-state MDP defined in Figure 1. Suppose again we have reward $r = 0$ for all state-action pairs. If we suppose that the dataset contains the sole transition $(s_0, a_0, r, s_1)$ then

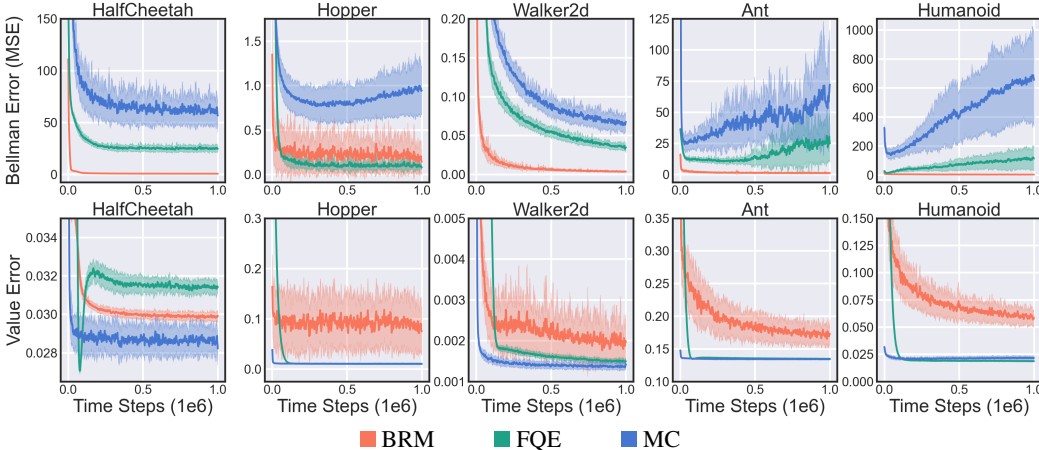

Figure 2: Comparing the Bellman error (top row) with value error (bottom row) of value functions trained with a dataset of 1m *on-policy* transitions. Error terms are evaluated over a held-out test set of on-policy rollouts. The shaded area captures the standard deviation over 10 seeds. MC refers to Monte Carlo value estimation with bootstrapping to reduce bias from time-delimited episode termination. While clearly the Bellman error is lowest for ■ BRM (which directly minimizes Bellman error) followed by ■ FQE (which indirectly minimizes Bellman error) followed by ■ MC (which minimizes the MC estimate of value error), this ordering is not reflected in value error. This shows overfitting of the Bellman error objective is possible even with on-policy data and that we cannot determine value prediction accuracy by examining empirical Bellman error alone.

we can construct examples where the Bellman error is 0 but the value error is arbitrarily large and conversely, where the Bellman error is arbitrarily large but the value error is 0.

**Example 1.** (0 **Bellman Error,** $C$ **Value Error**). We define the $Q$-values such that the Bellman error is 0 but the value error is $C$.

$$\text{If} \quad \begin{array}{l} Q(s_0, a_0) = C, \\ Q(s_1, a_0) = \frac{1}{\gamma}C. \end{array} \quad \implies \quad \begin{array}{l} \epsilon(s_0, a_0) = C - \gamma\frac{1}{\gamma}C = 0, \\ \Delta(s_0, a_0) = Q(s_0, a_0) - 0 = C. \end{array} \quad (7)$$

**Example 2.** ($C$ **Bellman Error,** 0 **Value Error**). In this second example, we define the $Q$-values such that the Bellman error is $C$ but the value error is 0.

$$\text{If} \quad \begin{array}{l} Q(s_0, a_0) = 0, \\ Q(s_1, a_0) = -\frac{1}{\gamma}C. \end{array} \quad \implies \quad \begin{array}{l} \epsilon(s_0, a_0) = 0 + \gamma\frac{1}{\gamma}C = C, \\ \Delta(s_0, a_0) = Q(s_0, a_0) - 0 = 0. \end{array} \quad (8)$$

Note that these examples do not involve adversarially modifying the environment in some extreme manner, and instead occur due to the value estimate of the missing transition. As a result, these scenarios can still happen in practical settings where function approximation is used to estimate the values of missing transitions, as it is difficult to control the behavior of function approximation and guarantee avoiding these scenarios where the Bellman error is deceptively low.

### 4.2 KEY EXPERIMENTS

Everything we have discussed thus far has suggested that Bellman error may not be a representative proxy objective for value error. We now examine our ideas with empirical results. Our main observation is that the relationship between Bellman error and value error is broken in finite data settings, particularly in the off-policy case. To do so, we examine the Bellman error and the value error of value functions trained by BRM and FQE. Additionally, we remark that our experiments are in deterministic domains, and as such, the problems we introduce are independent from the double sampling problem with BRM (Baird, 1995).

**On-policy empirical Bellman error is insufficient to rank value functions.** Figure 2 shows the learning curves of value functions trained with on-policy data, and evaluated on a held-out test set of on-policy rollouts. Recall that while FQE uses an iterative approach based on the Bellman equation, BRM directly minimizes the Bellman error, and the Monte Carlo estimate (MC) does not use the

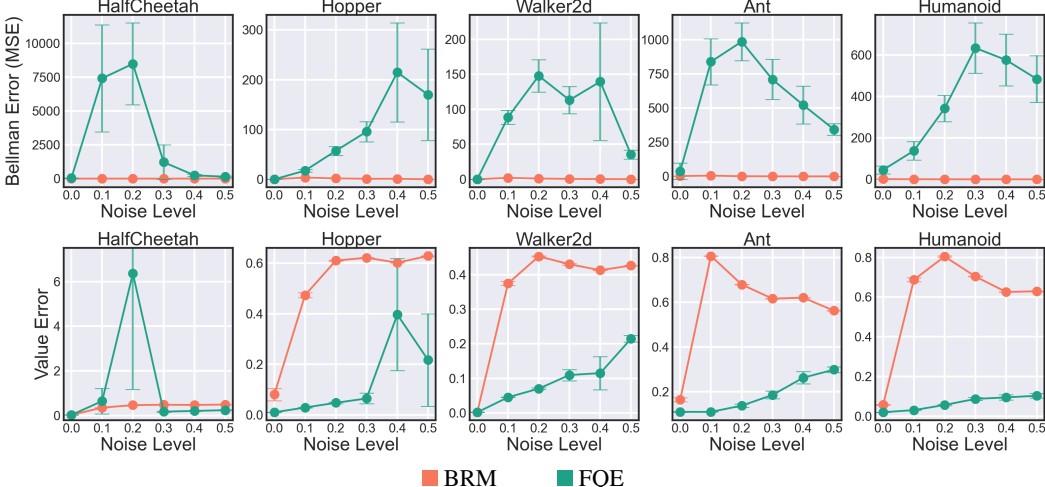

Figure 3: The final Bellman error and value error of functions trained with datasets gathered by increasingly noisy versions of the target policy. 0.0 is an on-policy dataset and the remainder are off-policy. Error bars capture the standard deviation over 10 seeds. Bellman error was clipped to 10k on the HalfCheetah task for FQE for visual clarity, as the value estimate diverged for 0.1 and 0.2 noise levels (see the learning curves in Appendix C.2). ■ FQE consistently outperforms ■ BRM while having significantly higher Bellman error. Additionally, while the value prediction accuracy of BRM drops substantially with increased distribution shift, the Bellman error term remains low throughout all settings, suggesting it is possible to train a function with low Bellman error, regardless of its value accuracy.

| Train Data | Test Data | Algorithm | HalfCheetah | Hopper | Walker2d | Ant | Humanoid |
|---|---|---|---|---|---|---|---|
| All | On-Policy | BRM | ■ 0.95 | ■ 0.74 | ■ 0.96 | ■ 0.99 | ■ 0.98 |
| | | FQE | ■ 0.81 | ■ 0.76 | ■ 0.72 | ■ 0.79 | ■ 0.11 |
| All | 0.1 | BRM | ■ -0.46 | ■ -0.83 | ■ -0.74 | ■ -0.75 | ■ -0.65 |
| | | FQE | ■ 0.57 | ■ 0.85 | ■ -0.90 | ■ -0.60 | ■ 0.20 |
| 0.1 | 0.1 | BRM | ■ 0.11 | ■ 0.04 | ■ -0.47 | ■ 0.46 | ■ -0.48 |
| | | FQE | ■ 0.92 | ■ 0.29 | ■ -0.14 | ■ 0.58 | ■ 0.05 |

Table 1: Pearson's correlation coefficient of the final Bellman error and value error of functions trained with either only BRM or only FQE. Warm colors ■ are used to show positive correlation and cold colors ■ are used for negative correlation. The error terms are computed over the test dataset. The functions are trained using datasets of varying noise levels, where all refers to the set (0.1, 0.2, 0.3, 0.4, 0.5) with 10 seeds, (6×10 functions), 0.1 refers to the subset of functions trained on the 0.1 dataset (10 functions). While there is high correlation between the on-policy empirical Bellman error and value error when comparing functions trained with the same algorithm, this relationship is not strong when evaluated with an off-policy dataset.

Bellman equation in its objective. Therefore, it is unsurprising that the value functions trained by BRM have lower Bellman error than the value functions trained by FQE and MC. However, even when the FQE value functions have much higher Bellman error (such as Ant and Humanoid), the results in value error have an inverse order, where the FQE and MC value functions have lower value error than the BRM value functions. These learning curves demonstrate that while BRM methods are capable of minimizing Bellman error more aggressively than FQE, the reduction in Bellman error is not reflected in value error. This means that even when working with large (1m) on-policy datasets, the empirical Bellman error should not be used to rank the performance of value functions.

Although Figure 2 shows a large variation in Bellman error, for most tasks, the value functions are competitive in terms of value prediction error. This means that although Bellman error is not an effective metric for selecting arbitrary value functions, it may still be effective as objective. In Figure 3 we test Bellman error as an off-policy objective. We display the final Bellman error and value error of value functions trained with off-policy datasets, where the data is gathered by a noisy version of the target policy. Next, we test Bellman error as a metric for value error across functions trained by a fixed algorithm. In Table 1, we seperate the functions by algorithm, and then measure the

| Train Data | Test Data | Metric | HalfCheetah | Hopper | Walker2d | Ant | Humanoid |
|---|---|---|---|---|---|---|---|
| All | On-Policy | BE | ■ 0.81 | ■ 0.76 | ■ 0.72 | ■ 0.79 | ▫ 0.11 |
| | | $\mathcal{L}_{\text{FQE}}$ | ■ 0.81 | ■ 0.79 | ■ 0.60 | ■ 0.81 | ▫ 0.22 |
| | | MSE | ■ 0.78 | ■ 0.95 | ■ 0.96 | ■ 0.98 | ■ 0.77 |
| All | 0.1 | BE | ■ 0.57 | ■ 0.85 | ■ -0.90 | ■ -0.60 | ▫ 0.20 |
| | | $\mathcal{L}_{\text{FQE}}$ | ■ 0.62 | ■ 0.84 | ■ -0.90 | ■ -0.59 | ▫ 0.21 |
| | | MSE | ■ 0.96 | ■ 0.84 | ■ 0.72 | ■ 0.97 | ■ 0.85 |

Table 2: Pearson's correlation coefficient of varying metrics and the value error of functions trained with. BE: Bellman Error, $\mathcal{L}_{\text{FQE}}$: the FQE objective, MSE: regression loss. This regression loss is with respect to a fixed target is taken from the FQE objective from a single trial (and then kept fixed across all trials). Warm colors ■ are used to show positive correlation and cold colors ■ are used for negative correlation. The error terms are evaluated over the test data. All functions (from Figure 3), trained with datasets of varying noise levels, are included. We can see that the difference between BE and $\mathcal{L}_{\text{FQE}}$ is minimal, and that MSE with a fixed target is the most effective metric. This shows that we cannot compare functions using $\mathcal{L}_{\text{FQE}}$ because it is non-stationary (and dependent on the current value function), but removing the dependency on the current value function (MSE) is a strong proxy for value error.

correlation between these error terms. Train data denotes the set of training data, i.e. 'All' includes the entire set of final functions gathered from the experiments in Figure 3, and '0.1' includes only the functions trained on the dataset of 0.1 noise level. The test data describes which dataset the error terms are evaluated on. Our conclusions are as follows:

**For a fixed algorithm, on-policy empirical Bellman error can be a good proxy for value error.** Table 1 shows that when we separate the value functions by algorithm, and evaluate the error terms over on-policy data, there is a strong correlation between the Bellman error and the value error. This means that Bellman error can be a meaningful learning objective when working with on-policy data, and explains the decent value prediction accuracy of BRM as shown in Figure 2.

**Bellman error is an ineffective off-policy objective.** The signal between Bellman error and value error is muddied when the error terms are evaluated with off-policy data (0.1). For BRM, we find the error terms correlate negatively. This is likely due to BRM overfitting to Bellman error objective. We also examine only the subset of functions which were trained with the 0.1 dataset, which means the functions compared are all trained with the same algorithm and on the same dataset. While they exhibit a higher correlation than set of functions trained with different datasets, the relationship is still not clear across all tasks. In Figure 3, we note that the value prediction of BRM degrades rapidly in the presence of distribution shift. Furthermore, in spite of this performance drop, the Bellman error of BRM value functions is low for all datasets, which shows that the empirical Bellman error can be optimized independently of value error, when evaluated over an off-policy dataset.

**BRM performs poorly because of premature convergence.** Our theoretical results, such as Corollary 1, show there exists infinitely many suboptimal solutions where the Bellman equation is satisfied. This means Bellman error is an unreliable signal for value error, as there are infinitely many functions with low Bellman error but high value error. However, in practice, we have found that the behavior of BRM is predictable. In Appendix A we examine the final values estimated by BRM, trained with data collected by different behavior policies of varying noise levels, and compare them to the true value of the target policy as well as the behavior policy. We find that BRM produces final values which are (1) highly consistent across seeds, (2) highly influenced by the behavior policy, and (3) close to 0. This means that BRM performs poorly as it tends to converge prematurely to the first low Bellman error solution (which is not reflected in low value error). Note that this premature convergence is not an aspect of FQE, which explains some of the performance gap between the two approaches. In the following section we will examine the success of FQE in more detail.

### 4.3 A MEANINGFUL OFF-POLICY BELLMAN OBJECTIVE REQUIRES GENERALIZATION

We now discuss the performance of FQE. In the previous section, we saw an evident disconnect Bellman error and value error. Our results show value functions trained with FQE can have growing Bellman error (Figure 2), and yet achieve a high value prediction accuracy (Figure 3). The success of FQE is supported by many examples in the literature for OPE tasks (Voloshin et al., 2019; Fu

et al., 2021; Fujimoto et al., 2021), as well as control applications with deep RL (Mnih et al., 2015; Lillicrap et al., 2015; Hessel et al., 2017). In this section, we discuss how FQE can be an effective approach, in spite of the flaws of the Bellman equation, and highlight the role of generalization in making the FQE objective a meaningful proxy for value error.

Unlike Bellman error, the objective used by FQE is dependent on a target value $\bar{Q}$:

$$\mathcal{L}_{\text{FQE}}(Q) := \frac{1}{|\mathcal{D}|} \sum_{(s,a,r,s') \sim \mathcal{D}} \left( Q(s,a) - (r + \gamma \bar{Q}(s',a')) \right)^2. \tag{9}$$

As FQE is an iterative algorithm, we can view $\mathcal{L}_{\text{FQE}}$ as an objective which is a function of the target $\bar{Q}$. Therefore, analysis of $\mathcal{L}_{\text{FQE}}$ will require reasoning about an inconsistent target. Instead, we might consider a fixed version of $\mathcal{L}_{\text{FQE}}$ where we use a single target across all trials. We should expect this metric to have increased relevance when comparing different value functions, as it is independent of the current value function. In similar fashion to Table 1, in Table 2, we list the correlation between value error and three metrics:

$$\text{Bellman error (BE)} : (Q_\theta(s,a) - (r + \gamma Q_\theta(s',a')))^2, \tag{10}$$

$$\text{The FQE objective } (\mathcal{L}_{\text{FQE}}) : (Q_\theta(s,a) - (r + \gamma Q_{\bar{\theta}}(s',a')))^2, \tag{11}$$

$$\text{Regression to a fixed target } \bar{Q} \text{ (MSE)} : (Q_\theta(s,a) - (r + \gamma \bar{Q}(s',a')))^2. \tag{12}$$

We use the subscript to show the parameters of the value function, where $\theta$ are the parameters of $Q$, $\bar{\theta}$ are the parameters of $\bar{Q}$ dependent on $\theta$, and $\bar{Q}$ is a fixed target network, independent of the $\theta$. We determine $\bar{Q}$ by taking the final target $Q_{\bar{\theta}}$ of a single trial, and then fix it across all trials.

The results in Table 2 show little difference between Bellman error and the FQE objective in determining value error. More importantly, we see MSE to a fixed target has a strong relationship to value error. Given we have previously shown that FQE learns an accurate value function (Figure 3), then it should be unsurprising that the distance to this target value function is a good proxy for value error. However, both the final Bellman error and FQE objective are *also* measuring the MSE to an accurate value function, but their values are dependent on the current value function, which makes them less valuable for comparing across different value functions. Ultimately, this experiment shows that the Bellman equation can be a useful off-policy objective if the target is accurate.

We can formalize the intuition *"if the target is accurate, then distance to the target is a good proxy for value error"* by the following proposition on proxy objectives.

**Proposition 3** *Given a pair value functions $(Q_1, Q_2)$ with value errors $(\Delta_1, \Delta_2)$, and target $y = r + \gamma \bar{Q}(s',a')$. If $sign(y - Q^\pi(s,a)) = sign(\Delta_1(s,a)) = sign(\Delta_2(s,a))$ and $|y - Q^\pi(s,a)| < \min(|\Delta_1(s,a)|, |\Delta_2(s,a)|)$ then $|Q_1(s,a) - y| < |Q_2(s,a) - y|$ implies $|\Delta_1(s,a)| < |\Delta_2(s,a)|$.*

This same observation could be applied equally to BRM methods. However, we remark that by optimizing both sides of the Bellman equation, BRM methods are directly modifying the target. We can understand some of the performance gap between FQE and BRM in that FQE is allowed to generalize, whereas in Figure 4, we see that BRM methods are pushed into early convergence, which favors solutions near 0, and inhibits generalization in the target.

**The Bellman equation needs generalization.** The importance of the accuracy of the target, highlights the reliance of the Bellman equation on generalization. With an off-policy and finite dataset, succeeding state-action pairs $(s', a')$ are unlikely to be contained in the dataset. Consequently, $Q(s', a')$ will only be accurate if the value function is able to generalize to this state-action pair[1].

While this is a simple observation, it has significant implications. Firstly, this means the Bellman equation requires generalization *during training*. This is distinct from typical machine learning settings, where generalization is an exercise which occurs after training. This is problematic because if it is difficult to ensure good generalization after training, it is only more difficult to ensure good generalization during training. This highlights the importance of feature learning (Jaderberg et al., 2016; Yang & Nachum, 2021), as neural network features are unlikely to be relevant early in training. Another implication is hyperparameter sensitivity. It is a well known problem that RL algorithms are sensitive to small adjustments (Henderson et al., 2017; Engstrom et al., 2019). A necessity of generalization at training time causes the significance of correct hyperparameters to be amplified.

---

[1]A similar observation has been made in the context of offline RL, with an emphasis on the *errors* this generalization induces (Fujimoto et al., 2019b).

## 5 RELATED WORK

The role of Bellman error has been considered in depth in the literature, in the context of bounds on the performance of a greedy policy in relation to the norm of the Bellman error (Williams & Baird, 1993; Singh & Yee, 1994; Bertsekas & Tsitsiklis, 1996; Heger, 1996; Munos, 2003; 2007; Farahmand et al., 2010).

Close to our work, Maillard et al. (2010) perform finite sample analysis on BRM methods with on-policy samples. Similar to our work, they conclude that the empirical Bellman error from on-policy samples is a reasonable approximation to the true Bellman error, but do not perform practical experiments or consider the off-policy setting. Kolter (2011) remarks that with off-policy sampling, the solution to linear TD can have arbitrarily large Bellman error but does not consider BRM methods, or finite datasets. Geist et al. (2017) evaluate the Bellman error as an objective for policy optimization. Although they examine a different setting, they arrive at a similar conclusion, the signal from the Bellman error is only meaningful if the sampling distribution corresponds to the optimal policy.

The Bellman error has additional concerns that our paper does not discuss. The double sampling problem (Baird, 1995) is that the gradient of the Bellman error is biased if estimated from a single transition in a stochastic MDP. The double sampling issue provides motivation for most recent BRM methods (Feng et al., 2019; Zhu & Ying, 2020; Bas-Serrano et al., 2020). We avoid this particular issue in our analysis by focusing on deterministic environments, but remark that BRM is likely to perform even worse with stochasticity. Sutton & Barto (1998) show that in scenarios where the feature representation of states is not uniquely defined, there exist examples where Bellman error is not learnable, as the structure of the MDP can not be determined from data alone, and thus the true Bellman error cannot be computed.

Our observations connect strongly to offline RL (Lange et al., 2012; Levine et al., 2020), where offline policy evaluation is used in conjunction with policy learning. Previous work has observed that the value function of FQE methods can diverge when computed offline due to poor estimates in the target (Fujimoto et al., 2018; 2019a; Kumar et al., 2019). Similar to our work, empirical properties of deep value functions which induce instability or divergence have been studied (Fu et al., 2019; Achiam et al., 2019) but have not considered the role of the objective itself. Several recent papers examined the sample complexity of offline RL, noting that without access to online data, the number of necessary transitions is exponential with respect to the horizon (Wang et al., 2020; Zanette, 2021; Chen et al., 2021; Xiao et al., 2021). In the context of offline model selection, several papers have observed that TD error is a weak baseline with poor correspondence to policy performance, remarking that it is a measure of value function accuracy rather than quality of the policy, but provide little analysis (Irpan et al., 2019; Paine et al., 2020; Tang & Wiens, 2021). Our findings help explain these results by showing that Bellman (and TD) error are not an effective measure of value accuracy, and cannot rank models even with on-policy data. These empirically-minded observations caution against traditional results which suggest Bellman error as a metric for model selection (Farahmand & Szepesvári, 2011).

## 6 CONCLUSION

In this paper we examine the role of the Bellman equation as an objective. Our main observation is that the Bellman equation is only uniquely solved by the true value function when computed over the entire MDP. For a given finite dataset, we show there can exist infinitely many suboptimal value functions which satisfy the Bellman equation. This exposes a fundamental problem with Bellman error, in that it is not guaranteed to correspond to value error. We demonstrate this problem theoretically, with toy problems, and empirically on standard benchmark environments. This result is highlighted by an empirical comparison between Bellman Residual Minimization (BRM) (Baird, 1995) and Fitted Q-Evaluation (FQE) (Ernst et al., 2005; Le et al., 2019), which shows that value functions trained with BRM have much lower Bellman error but much higher value error than value functions trained with FQE. While much of the modern literature surrounding Bellman error minimization emphasizes the double sampling problem (Dai et al., 2018; Feng et al., 2019; Saleh & Jiang, 2019; Bas-Serrano et al., 2020), our results show a much more fundamental problem; *solving the Bellman equation over a finite dataset does not guarantee an accurate value function*. We give concrete evidence of this problem with practical experiments, and hope our findings provide practitioners, and theorists alike, a better understanding of Bellman equation-based objectives.

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

# A    BRM FINAL VALUES

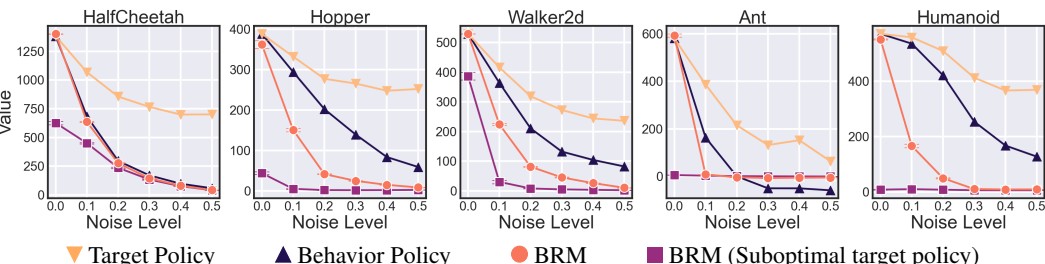

Figure 4: Visualizing the final value estimated by BRM after training on different datasets corresponding to varying noise levels. The true value of the target policy and the behavior policy are displayed to provide reference, as well as BRM when trained to evaluate a suboptimal policy (corresponding to TD3 trained for 300k time steps rather than 3m). Error bars capture the standard deviation over 10 seeds (but are visually hard to see as the deviation is low). We can see that ● BRM typically converges to a value which is closer to the ▲ behavior policy rather than the ▼ target policy, and typically prefers values which are close to 0. Interestingly, the BRM trained to evaluate the suboptimal target policy converges to the same value on the noisiest datasets, suggesting that the influence of the target policy on what value BRM converges to is reduced with increased distribution shift.

# B  PROOFS

## B.1  PROPOSITION 1

**Proposition 1** *If the Bellman error $\epsilon(s, a) = 0$ for all state-action pairs $(s, a) \in \mathcal{S} \times \mathcal{A}$, then the value error $\Delta(s, a) = 0$ for all state-action pairs $(s, a) \in \mathcal{S} \times \mathcal{A}$.*

*Proof.* This is a direct consequence of Theorem 1.

∎

## B.2  PROPOSITION 2

**Proposition 2** *For any constant $C > 0$ and discount factor $\gamma \in (0, 1)$, there exists an MDP and a pair of value functions $(Q_1, Q_2)$ with Bellman errors $(\epsilon_1, \epsilon_2)$ and value errors $(\Delta_1, \Delta_2)$, such that for all state-action pairs $(s, a) \in \mathcal{S} \times \mathcal{A}$, the absolute Bellman error of $Q_2$ is greater than the absolute Bellman error of $Q_1$ by $C$, $|\epsilon_2(s, a)| - |\epsilon_1(s, a)| > C$, but the absolute value error of $Q_1$ is greater than the absolute value error of $Q_2$ by $C$, $|\Delta_1(s, a)| - |\Delta_2(s, a)| > C$.*

*Proof.* Proof by construction. Consider an infinitely long chain MDP, with reward $r = 0$ for all transitions. We will consider a more general case, where $|\epsilon_2(s, a)|^b - |\epsilon_1(s, a)|^b > C$ and $|\Delta_1(s, a)|^d - |\Delta_2(s, a)|^d > C$, for any $b > 0$ and $d > 0$.

Let $k > 0$. Let $Q_1(\cdot) = \frac{k}{1-\gamma}$ for all inputs, and let $Q_2(s_t) = (-1)^t k$, in other words, $k$ on even states and $-k$ on odd states.

Since the value of all state-action pairs is 0, we have

$$|\Delta_1(\cdot)|^d = \left( \frac{k}{1-\gamma} \right)^d, \tag{13}$$

$$|\Delta_2(\cdot)|^d = k^d, \tag{14}$$

and

$$|\epsilon_1(\cdot)|^b = \left| \frac{k}{1-\gamma} - \frac{\gamma k}{1-\gamma} \right|^b = k^b, \tag{15}$$

$$|\epsilon_2(\cdot)|^b = |\pm 1(k - -\gamma k)|^b = (k + \gamma k)^b. \tag{16}$$

We have $|\Delta_1(s, a)|^d - |\Delta_2(s, a)|^d > C$ if:

$$\left( \frac{k}{1-\gamma} \right)^d - k^d > C \tag{17}$$

$$\iff \frac{1}{(1-\gamma)^d} k^d - k^d > C \tag{18}$$

$$\iff \left( \frac{1}{(1-\gamma)^d} - 1 \right) k^d > C \tag{19}$$

$$\iff k > \left( \frac{C}{\frac{1}{(1-\gamma)^d} - 1} \right)^{\frac{1}{d}}. \tag{20}$$

We have $|\epsilon_2(s, a)|^b - |\epsilon_1(s, a)|^b > C$ if:

$$(k + \gamma k)^b - k^b > C \tag{21}$$

$$\iff (1 + \gamma)^b k^b - k^b > C \tag{22}$$

$$\iff ((1 + \gamma)^b - 1) k^b > C \tag{23}$$

$$\iff k > \left( \frac{C}{(1+\gamma)^b - 1} \right)^{\frac{1}{b}}. \tag{24}$$

$$\tag{25}$$

Let $k = \max\left( \left( \frac{C}{(1+\gamma)^b - 1} \right)^{\frac{1}{b}}, \left( \frac{C}{\frac{1}{(1-\gamma)^d} - 1} \right)^{\frac{1}{d}} \right)$.

■

### B.3 THEOREM 1

**Theorem 1** *Let $d^\pi(s', a'|s, a) = (1 - \gamma) \sum_{t=0}^{\infty} \gamma^t p^\pi((s, a) \to s', t)\pi(a'|s')$, be the conditional discounted state-action occupancy, where $p^\pi((s, a) \to s, t)$ is the probability of leaving the state-action pair $(s, a)$ and visiting the state $s$ after $t$ time steps. The value error $\Delta(s, a)$ of a state-action pair $(s, a)$ can be defined as a function of the Bellman error $\epsilon(s', a')$ over $d^\pi(s', a'|s, a)$:*

$$\Delta(s, a) = \frac{1}{1 - \gamma} \mathbb{E}_{(s', a') \sim d^\pi(\cdot|s, a)}[\epsilon(s', a')]. \tag{26}$$

*Proof.* Our proof follows similar steps to the proof of Lemma 6.1 in (Kakade & Langford, 2002) and likely others.

First by definition:

$$\Delta(s, a) := Q(s, a) - Q^\pi(s, a) \tag{27}$$
$$\Rightarrow Q^\pi(s, a) = Q(s, a) - \Delta(s, a). \tag{28}$$

Then we can decompose value error:

$$\Delta(s, a) = Q(s, a) - Q^\pi(s, a) \tag{29}$$
$$= Q(s, a) - (r + \gamma \mathbb{E}_\pi[Q^\pi(s', a')]) \tag{30}$$
$$= Q(s, a) - (r + \gamma \mathbb{E}_\pi[Q(s', a') - \Delta(s', a')]) \tag{31}$$
$$= Q(s, a) - (r + \gamma \mathbb{E}_\pi[Q(s', a')]) + \gamma \mathbb{E}_\pi[\Delta(s', a')] \tag{32}$$
$$= \epsilon(s, a) + \gamma \mathbb{E}_\pi[\Delta(s', a')]. \tag{33}$$

By treating $\Delta(s, a)$ as a value function and $\epsilon(s', a')$ as the reward, we can see that:

$$\Delta(s, a) = \frac{1}{1 - \gamma} \mathbb{E}_{(s', a') \sim d^\pi(\cdot|s, a)}[\epsilon(s', a')]. \tag{34}$$

Note that this theorem can also be applied to finite horizon MDPs, by either considering a definition of $d^\pi$ which accounts for the finite horizon, $d^\pi(s', a'|s, a) = \frac{1}{\sum_{t=0}^{T-1} \gamma^t} \sum_{t=0}^{T-1} \gamma^t p^\pi((s, a) \to s', t)\pi(a'|s')$, or by transforming the finite horizon MDP into an infinite horizon MDP by considering episode termination to be a terminal state which loops infinitely upon itself.

■

### B.4 COROLLARY 1

**Corollary 1** *If there exists a state-action pair $(s', a')$ not contained in the dataset $\mathcal{D}$, where the state-action occupancy $d^\pi(s', a'|s, a) > 0$, then for any $C > 0$, there exists a value function such that the Bellman error is $0$ for all state-action pairs in the dataset $\mathcal{D}$, while the value error of the state-action pair $(s, a)$ is $C$.*

*Proof.* This is a direct consequence of Theorem 1. Let $\mathcal{D}'$ contain the set of state-action pairs $(s', a')$ not contained in the dataset $\mathcal{D}$, where the state-action occupancy $d^\pi(s', a'|s, a) > 0$. It follows that

$$\Delta(s, a) = \frac{1}{1 - \gamma} \mathbb{E}_{(s', a') \sim d^\pi(\cdot|s, a)}[\epsilon(s', a')] \tag{35}$$

$$= \frac{1}{1 - \gamma} \sum_{(s', a') \sim \mathcal{D}} \epsilon(s', a') + \frac{1}{1 - \gamma} \sum_{(s', a') \sim \mathcal{D}'} \epsilon(s', a'). \tag{36}$$

Recall

$$\epsilon(s,a) := Q(s,a) - \mathbb{E}_{r,s'\sim p, a'\sim\pi}\left[r + \gamma Q(s',a')\right], \tag{37}$$

and there exists at least one $Q(s,a)$, such that $(s,a) \in \mathcal{D}'$. It follows that we can choose a function $Q$ such that $\frac{1}{1-\gamma}\sum_{(s',a')\sim\mathcal{D}} \epsilon(s',a') = 0$ but $\frac{1}{1-\gamma}\sum_{(s',a')\sim\mathcal{D}'} \epsilon(s',a') = C$.

■

## B.5 PROPOSITION 3

**Proposition 3** *Given a pair value functions $(Q_1, Q_2)$ with value errors $(\Delta_1, \Delta_2)$, and target $y = r + \gamma\bar{Q}(s',a')$. If $sign(y - Q^\pi(s,a)) = sign(\Delta_1(s,a)) = sign(\Delta_2(s,a))$ and $|y - Q^\pi(s,a)| < \min(|\Delta_1(s,a)|, |\Delta_2(s,a)|)$ then $|Q_1(s,a) - y| < |Q_2(s,a) - y|$ implies $|\Delta_1(s,a)| < |\Delta_2(s,a)|$.*

$$|Q_1(s,a) - y| < |Q_2(s,a) - y| \tag{38}$$
$$\Rightarrow |Q_1(s,a) - Q^\pi(s,a) + Q^\pi(s,a) - y| < |Q_2(s,a) - Q^\pi(s,a) + Q^\pi(s,a) - y| \tag{39}$$
$$\Rightarrow |\Delta_1(s,a) - (y - Q^\pi(s,a))| < |\Delta_2(s,a) - (y - Q^\pi(s,a))| \tag{40}$$
$$\Rightarrow |\Delta_1(s,a)| - |(y - Q^\pi(s,a))| < |\Delta_2(s,a)| - |(y - Q^\pi(s,a))| \tag{41}$$
$$\Rightarrow |\Delta_1(s,a)| < |\Delta_2(s,a)|. \tag{42}$$

Where the second last line is from $sign(y - Q^\pi(s,a)) = sign(\Delta_1(s,a)) = sign(\Delta_2(s,a))$ and $|y - Q^\pi(s,a)| < \min(|\Delta_1(s,a)|, |\Delta_2(s,a)|)$.

■

# C LEARNING CURVES

## C.1 BRM

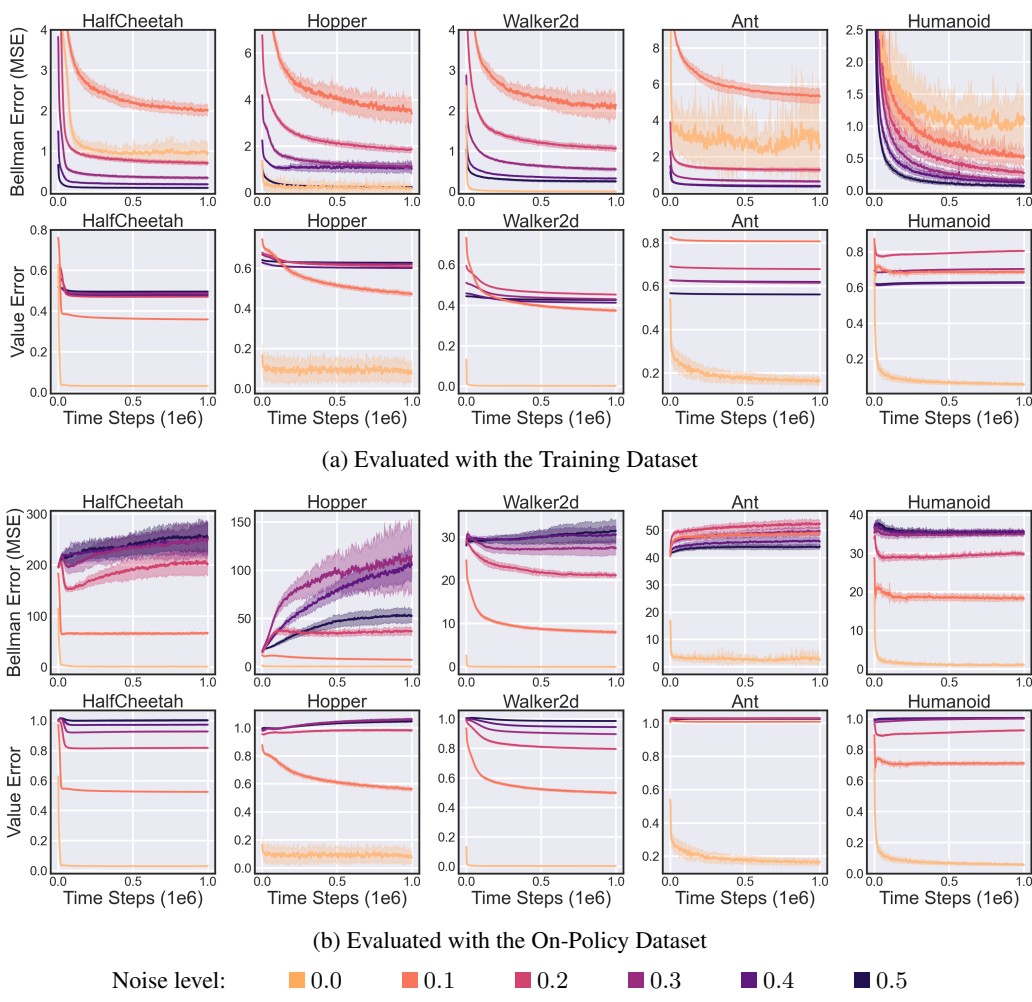

(a) Evaluated with the Training Dataset

(b) Evaluated with the On-Policy Dataset

Noise level: ■ 0.0 ■ 0.1 ■ 0.2 ■ 0.3 ■ 0.4 ■ 0.5

Figure 5: Visualizing the learning curves of the Bellman error and value error of value functions trained by BRM. The shaded area captures the standard deviation over 10 seeds. We can observe that Bellman error evaluated with the on-policy dataset is roughly representative of the value error, while the Bellman error evaluated with their training datasets is not.

## C.2 FQE

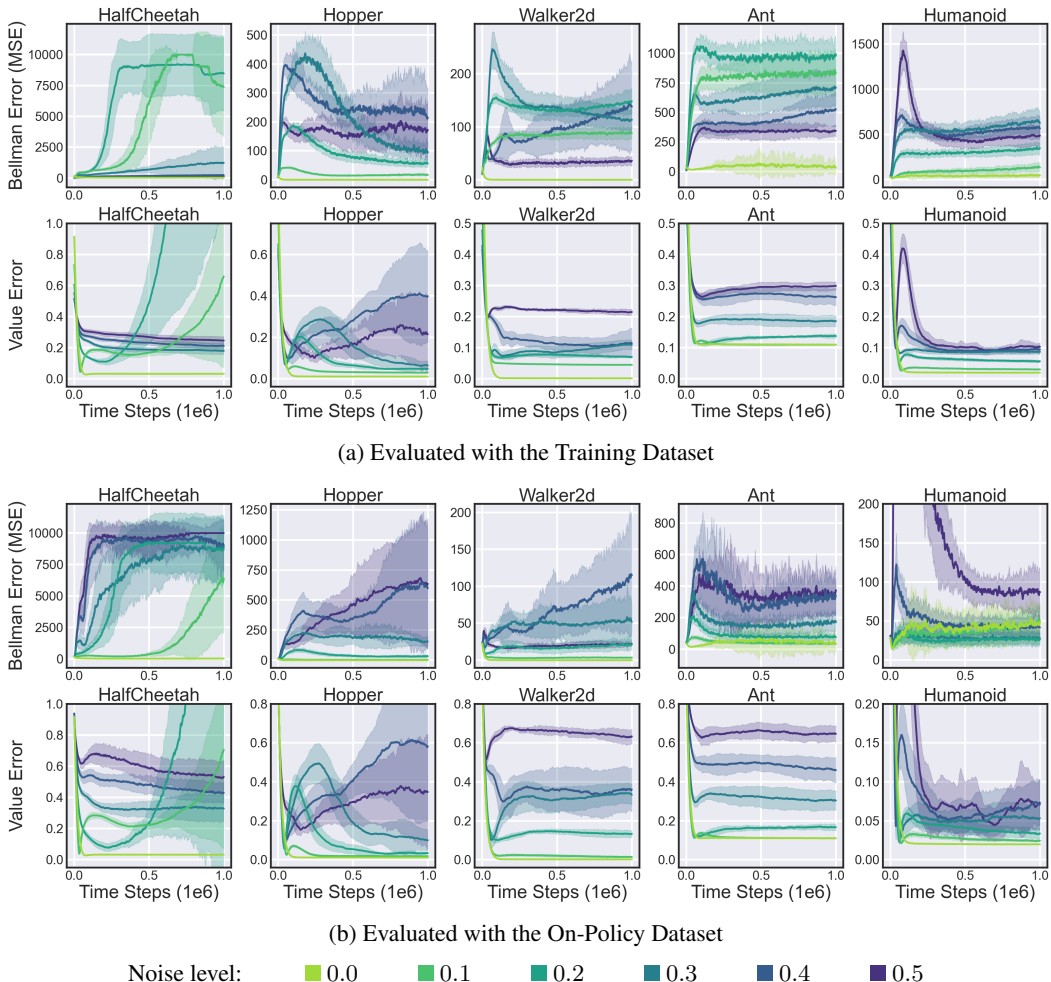

(a) Evaluated with the Training Dataset

(b) Evaluated with the On-Policy Dataset

Noise level: ■ 0.0 ■ 0.1 ■ 0.2 ■ 0.3 ■ 0.4 ■ 0.5

Figure 6: Visualizing the learning curves of the Bellman error and value error of value functions trained by FQE. The shaded area captures the standard deviation over 10 seeds. Bellman error of individual trials is clipped to 10k for visual clarity.

# D WHAT IF...

## D.1 ...WE COMPARE ABSOLUTE BELLMAN ERROR TO ABSOLUTE VALUE ERROR?

Figure 2 and Table 1 compare the mean squared Bellman error to the absolute value error. In this section we repeat these results comparing absolute Bellman error to absolute value error (Figure 7, Table 3), and mean squared Bellman error to mean squared value error (Figure 8, Table 4). We find that our observations are consistent with the observations in the main body.

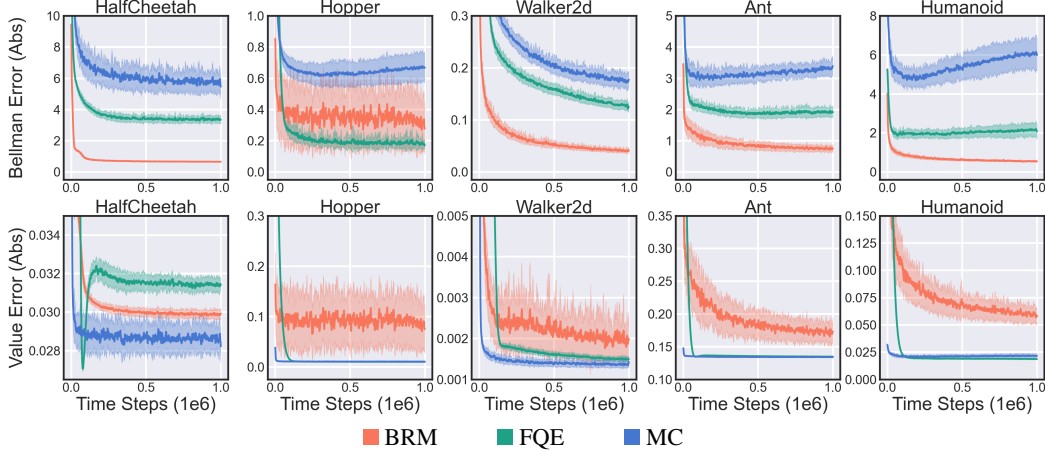

Figure 7: Comparing the absolute Bellman error with the absolute value error. The shaded area captures the standard deviation over 10 seeds. Both algorithms are trained using on-policy data collected by the target policy.

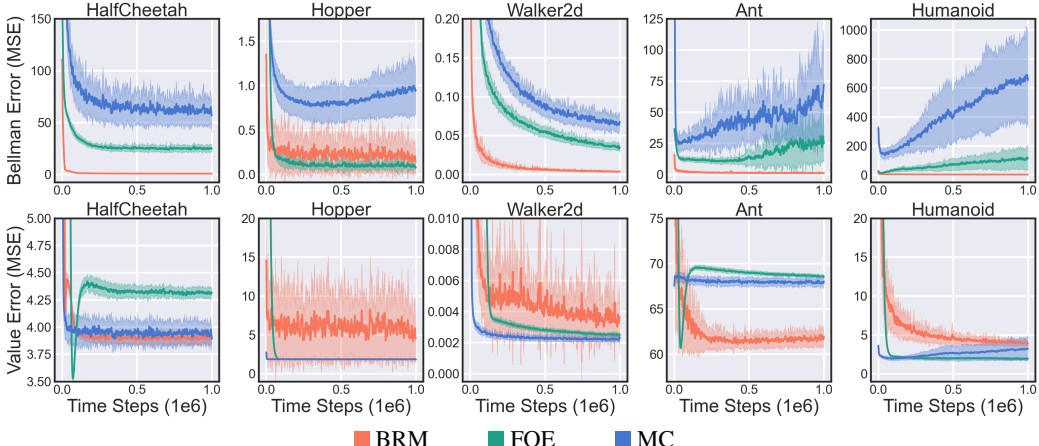

Figure 8: Comparing the mean squared Bellman error with the mean squared value error. The shaded area captures the standard deviation over 10 seeds. Both algorithms are trained using on-policy data collected by the target policy.

| Train Data | Test Data | Algorithm | HalfCheetah | Hopper | Walker2d | Ant | Humanoid |
|---|---|---|---|---|---|---|---|
| All | On-Policy | BRM | ■ 1.00 | ■ 0.97 | ■ 1.00 | ■ 1.00 | ■ 1.00 |
|  |  | FQE | ■ 0.79 | ■ 0.79 | ■ 0.80 | ■ 0.79 | ■ 0.83 |
| All | 0.1 | BRM | 0.01 | ■ 0.72 | ■ -0.18 | ■ -0.61 | ■ -0.50 |
|  |  | FQE | ■ 0.73 | ■ 0.54 | ■ -0.58 | -0.07 | ■ 0.67 |
| 0.1 | 0.1 | BRM | -0.08 | ■ 0.53 | -0.14 | ■ 0.49 | -0.15 |
|  |  | FQE | ■ 0.99 | ■ 0.76 | 0.01 | ■ 0.43 | ■ 0.27 |

Table 3: Pearson's correlation coefficient of the final absolute Bellman error and absolute value error of functions trained with either only BRM or only FQE. Warm colors ■ are used to show positive correlation and cold colors ■ are used for negative correlation. The error terms are computed over the test dataset. The functions are trained using datasets of varying noise levels, where all refers to the set (0.1, 0.2, 0.3, 0.4, 0.5) with 10 seeds, (6×10 functions), 0.1 refers to the subset of functions trained on the 0.1 dataset (10 functions). This is a repeat of Table 1, comparing absolute Bellman error to absolute value error (rather than MSE Bellman error). Similar observations can be made.

| Train Data | Test Data | Algorithm | HalfCheetah | Hopper | Walker2d | Ant | Humanoid |
|---|---|---|---|---|---|---|---|
| All | On-Policy | BRM | ■ 0.97 | ■ 0.79 | ■ 0.99 | ■ 0.99 | ■ 1.00 |
|  |  | FQE | ■ 0.63 | ■ 0.63 | ■ 0.60 | ■ 0.55 | ■ 0.15 |
| All | 0.1 | BRM | ■ -0.29 | ■ -0.70 | ■ -0.58 | ■ -0.77 | ■ -0.58 |
|  |  | FQE | ■ 0.56 | ■ 0.82 | ■ -0.91 | ■ -0.66 | ■ 0.20 |
| 0.1 | 0.1 | BRM | ■ 0.95 | ■ 0.32 | -0.13 | ■ 0.53 | ■ 0.54 |
|  |  | FQE | 0.12 | 0.06 | ■ -0.47 | ■ 0.28 | ■ -0.48 |

Table 4: Pearson's correlation coefficient of the final mean squared Bellman error and mean squared value error of functions trained with either only BRM or only FQE. Warm colors ■ are used to show positive correlation and cold colors ■ are used for negative correlation. The error terms are computed over the test dataset. The functions are trained using datasets of varying noise levels, where all refers to the set (0.1, 0.2, 0.3, 0.4, 0.5) with 10 seeds, (6×10 functions), 0.1 refers to the subset of functions trained on the 0.1 dataset (10 functions). This is a repeat of Table 1, comparing mean squared Bellman error to mean squared value error (rather than absolute value error). Similar observations can be made.

## D.2 …WE USE LESS DATA?

All of our experiments use datasets of 1m transitions. In Figure 9 we repeat the experiment in Figure 2 with a dataset of 50k transitions, rather 1m. We find our observations are unchanged in this setting, in that FQE and MC have much higher Bellman errors but lower value errors than BRM.

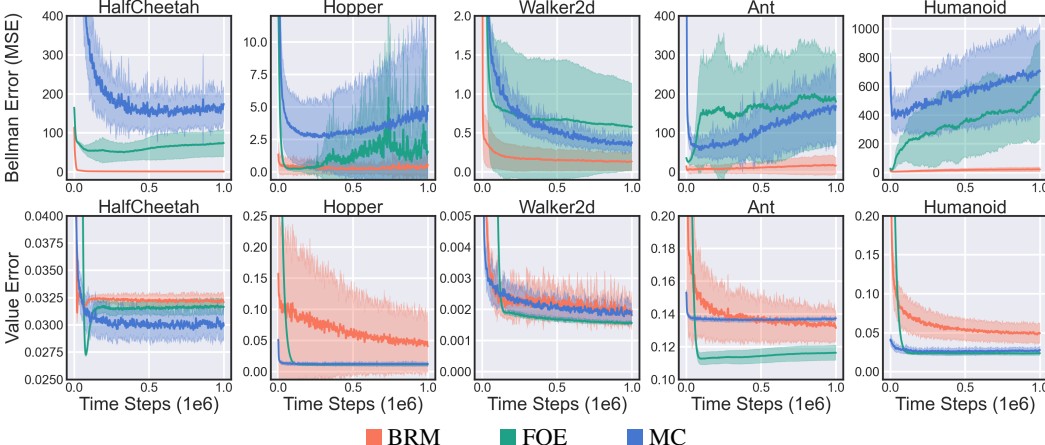

Figure 9: Comparing the mean squared Bellman error with the absolute value error, using a dataset of 50k, rather than 1m as in Figure 2. The shaded area captures the standard deviation over 10 seeds. Both algorithms are trained using on-policy data collected by the target policy.

### D.3 …WE USE TRAINING ERROR INSTEAD OF TEST ERROR?

Figure 2 evaluates the Bellman error and the value error on a held-out test set. In section we show the results on the training set (Figure 10). We see near identical figures. This shows that the dataset is sufficiently large that overfitting to individual transitions does not occur.

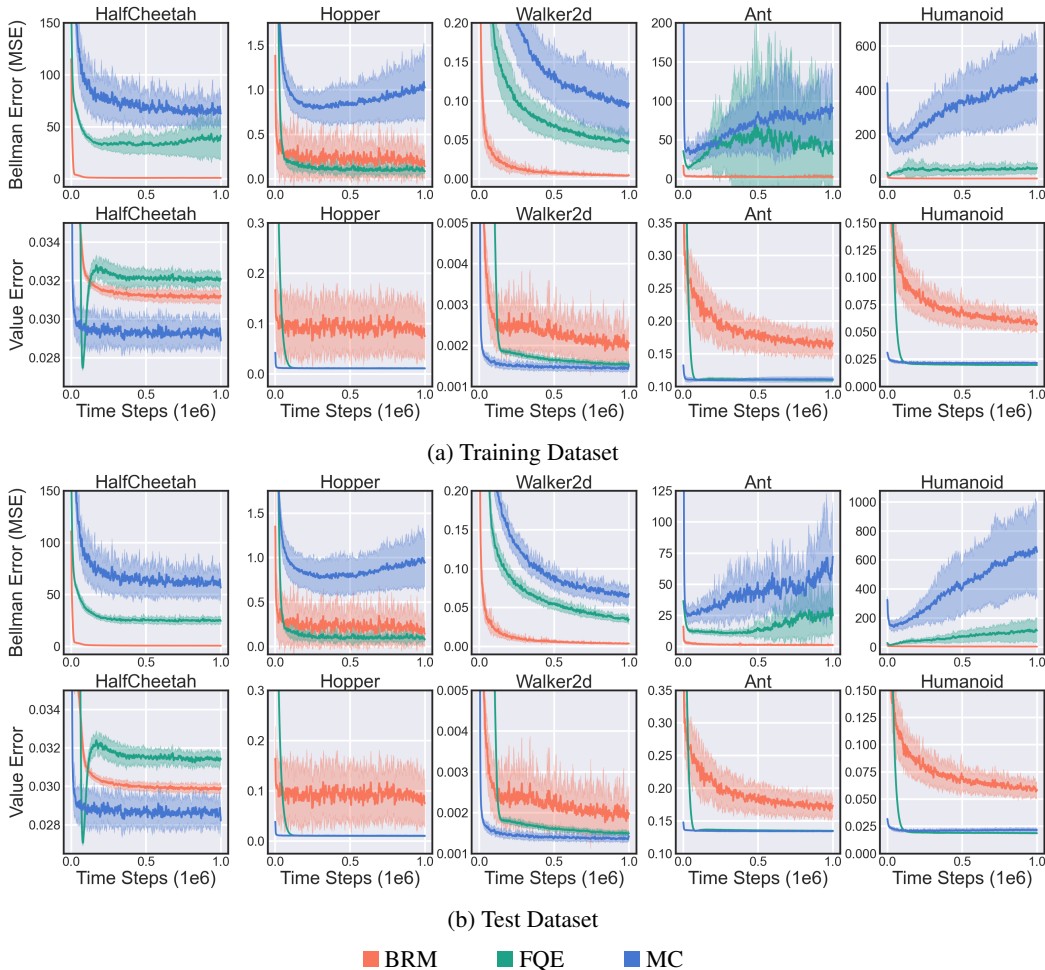

Figure 10: Comparing the Bellman error with value error. The shaded area captures the standard deviation over 10 seeds. Both algorithms are trained using on-policy data collected by the target policy. Figure 2 shows the Bellman error and value error, evaluated on a held-out test set (repeated in Figure 10b). In this figure we also display the error terms over the training set (Figure 10a). We see near identical figures. This shows that the dataset is sufficiently large that overfitting to individual transitions does not occur.

### D.4   ...WE USE TRAIN FOR LONGER?

All of our experiments train for 1m timseteps. In Figure 11 we repeat the experiment in Figure 2 while training for 10m time steps, rather 1m. Interestingly, we see more consistent relative ordering between FQE and BRM at 10m time steps, but otherwise our observations are unchanged with additional training.

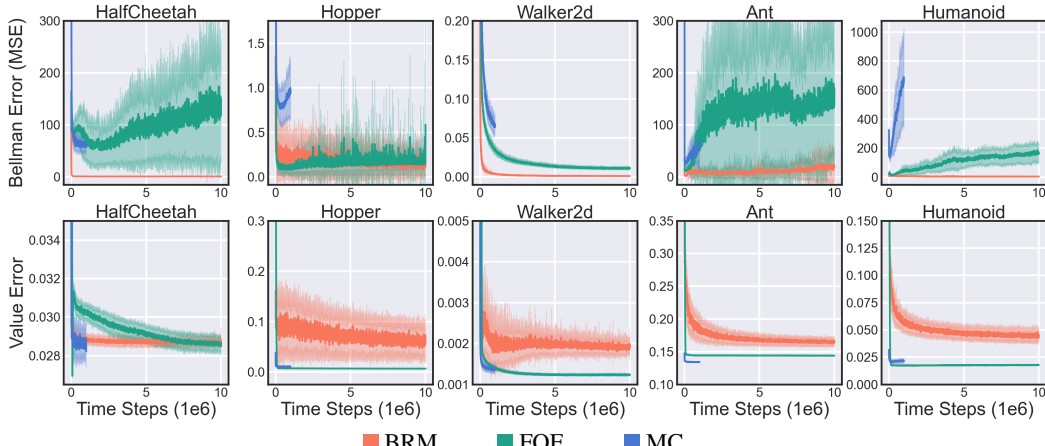

Figure 11: Comparing the mean squared Bellman error with the absolute value error using a dataset of 1m but trained for 10m time steps total, rather than 1m as in Figure 2. The shaded area captures the standard deviation over 10 seeds. Both algorithms are trained using on-policy data collected by the target policy.

# E  EXPERIMENT DETAILS

**Software.** Software versions used were as follows:

- Python 3.6
- Pytorch 1.8.0 (Paszke et al., 2019)
- Gym 0.17.0 (Brockman et al., 2016)
- MuJoCo 1.50[2] (Todorov et al., 2012)
- mujoco-py 1.50.1.1

-v3 versions of the MuJoCo environments were used.

**Hyperparameters.** FQE and BRM used the same hyperparameters and architecture, as described in Table 5. These hyperparameters were chosen to match TD3 and SAC (Fujimoto et al., 2018; Haarnoja et al., 2018a), state of the art off-policy RL algorithms used in the MuJoCo domain. Following these algorithms, both FQE and BRM set the discount factor $\gamma$ to 0 for terminal states (and use 0.99 otherwise). FQE uses Polyak averaging for the target network update. Given parameters $\theta$ of the current network, the parameters of the target network $\bar{\theta}$ are updated by the following after each time step:

$$\bar{\theta} \leftarrow (1 - \tau)\bar{\theta} + \tau\theta, \tag{43}$$

where $\tau$ is the target update rate. This rule is a commonly-used update rule by many off-policy RL algorithms for continuous actions (Lillicrap et al., 2015; Fujimoto et al., 2018; Haarnoja et al., 2018b).

| | Hyperparameter | Value |
|---|---|---|
| Network Hyperparameters | Optimizer | Adam (Kingma & Ba, 2014) |
| | Learning rate | 3e-4 |
| | Mini-batch size | 256 |
| | Target update rate (FQE) | 5e-3 |
| | Discount factor | 0.99 |
| | Terminal Discount factor | 0.0 |
| Architecture | Network Hidden dim | 256 |
| | Network Hidden layers | 2 |
| | Activation function | ReLU |

Table 5: Hyperparameters and architecture.

**OPE target.** In each experiment, we evaluate the discounted return of a deterministic target policy taken from TD3 (Fujimoto et al., 2018) trained for 3 million time steps. Our implementation of TD3 is based directly off of the author-provided Github (https://github.com/sfujim/TD3). For all experiments, the discounted return uses a discount factor of $\gamma = 0.99$.

**Dataset and behavior policy.** Datasets are collected by using a noisy variation of the target policy $\pi_t$. For a noise level $n \in [0.0, 0.1, 0.2, 0.3, 0.4, 0.5]$, the behavior policy $\pi_b$ is defined as:

$$\pi_b(s) = \begin{cases} \pi_t(s) + \mathcal{N}(0, n), & \text{with } p = n, \\ \text{uniform random action} & \text{with } p = 1 - n. \end{cases} \tag{44}$$

Most experiments use a dataset of 1 million transitions, matching the replay buffer size of TD3/SAC.

**Metrics and evaluation datasets.** The main metrics used are Bellman error and value error. Given an evaluation dataset $\mathcal{D}_e$ and value function $Q$, the mean-squared Bellman error is computed by:

$$\frac{1}{|\mathcal{D}_e|} \sum_{(s,a,r,s') \sim \mathcal{D}_e, a' \sim \pi} \left(Q(s,a) - (r + \gamma Q(s', a'))\right)^2. \tag{45}$$

Given an evaluation dataset $\mathcal{D}_e$ and value function $Q$, the normalized absolute value error is computed by:

$$\frac{1}{K|\mathcal{D}_e|} \sum_{(s,a,r,s') \sim \mathcal{D}_e, a' \sim \pi} |Q(s,a) - Q^\pi(s,a)|. \tag{46}$$

---

[2]License information: https://www.roboti.us/license.html

$Q^\pi(s, a)$ is computed near exactly by resetting the MuJoCo environment to the specific state-action pair $(s, a)$ and running the policy for 1000 time steps. Value error is normalized by a per-environment constant equal to the average true value $Q^\pi$ over the on-policy evaluation dataset $\mathcal{D}_{0.0}$:

$$K = \frac{1}{|\mathcal{D}_{0.0}|} \sum_{(s,a,r,s') \sim \mathcal{D}_{0.0}} Q^\pi(s, a). \tag{47}$$

We report the values of $K$ used in Table 6.

| Environment | $K$ |
|---|---|
| HalfCheetah | 1382.35 |
| Hopper | 388.56 |
| Walker2d | 529.12 |
| Ant | 580.90 |
| Humanoid | 571.29 |

Table 6: Values of the per-environment normalizing constant $K$, used to normalize value error for better interpretability across tasks.

Evaluation datasets are each collected by using the same set of behavior policies used to generate the training datasets, in other words with noise levels $[0.0, 0.1, 0.2, 0.3, 0.4, 0.5]$. Each evaluation dataset contains 1000 transitions, and is gathered by collecting 50k transitions, and then uniformly randomly saving 1000 of the 50k transitions. Error terms are computed over an evaluation dataset of 1000 transitions, generated in similar fashion as the training datasets. Tables (1 & 2) report Pearson's correlation coefficient. Since this measure is not robust to outliers, for FQE we remove the 30% of data points with the highest Bellman error terms (functions trained with BRM had no obvious outliers).

