# OpenReview forum: "Why Should I Trust You, Bellman? Evaluating the Bellman Objective with Off-Policy Data"
_ICLR.cc/2022/Conference — ICLR 2022 Submitted_

### Official Review · Reviewer_fSvY · 2021-10-23

**Correctness:** 4
**Technical Novelty And Significance:** 3
**Empirical Novelty And Significance:** 3
**Recommendation:** 6
**Confidence:** 4

**Main Review:**

Strength:
The message sent by this paper is very interesting and important. This paper challenges one broadly accepted method of using Bellman error as the objective function. It proves that minimizing empirical Bellman error can be far from minimizing the value error, especially for the off-policy evaluation problems. Their theoretical results are corroborated by experimental evidence.

Weakness:
The theoretical results of the paper are not deep enough It basically only gives some existence result which is good but not helpful enough. It actually feels a little bit obvious.
Besides, the case considered in the paper are limited. It only considers the discounted MDP setting, while ignoring many other important settings such as finite-horizon MDPs and goal-oriented MDP (i.e. shortest path). Therefore I wonder if similar results can hold for a broader class of models/problems.


**Summary Of The Paper:**

This paper studies the relation between the Bellman equation and the accuracy of the value functions. This paper shows that Bellman error is not a good measure for comparing value functions and thus is an ineffective objective function for off-policy evaluation.


**Summary Of The Review:**

Despite the weakness mentioned above, I still think this paper is good. The message is important and the results are novel and interesting.

My current recommendation is (weak) accept.

---

> ### Author Response · Authors · 2021-11-17
> **Response to Reviewer fSvY**
>
> Thank you for your comments! With regards to the theoretical results, we feel as if this paper is more of an empirical evaluation, and thus existence results followed by exhaustive empirical evidence should well justify our arguments.
>
> Regarding settings, the Bellman equation (and most of our main results) naturally extend to the finite horizon/episodic setting by considering a terminal state at the end of each episode (which loops to itself infinitely). We’ve added some comments which extend Theorem 1 (and therefore Corollary 1) to the finite horizon setting in the appendix. We also remark that goal-oriented MDPs are a subset of finite horizon MDPs, so this covers that setting as well.
>
> Please let us know if you feel there are any other ways to improve the paper. Thank you.

---

### Official Review · Reviewer_oTrW · 2021-10-24

**Correctness:** 4
**Technical Novelty And Significance:** 3
**Empirical Novelty And Significance:** Not applicable
**Recommendation:** 8
**Confidence:** 4

**Main Review:**

### **Strengths**
+ Exposition is clear and easy to follow.
+ The paper includes theoretical analyses that are simple and to the point, and makes good use of illustrative examples.
+ Empirical experiments test various settings to support the main claims, and results are nicely summarized in figures/tables.

### **Weaknesses**
- Could you comment on the connection between the notion of “generalizability during training” and “realizability/completeness of function approximator” [1]
- There are two ways Bellman error can be used: (i) as the loss function to optimize, and (ii) as a metric to compare two different value functions. The authors provided a good review of (i) in Sec 5, which is more on the learning aspect. More recently, researchers looking at the model selection aspect, which is more related to (ii), have made similar observations that the TD error loss does not correlate well with policy value/performance [2][3][4]. Therefore, I think adding a discussion on model selection can strengthen the work and potentially resonate with more researchers.


### **References**
1. Jinglin Chen, Nan Jiang. Information-Theoretic Considerations in Batch Reinforcement Learning. ICML 2019. https://arxiv.org/abs/1905.00360
2. Tom Le Paine et al. Hyperparameter Selection for Offline Reinforcement Learning. 2020. https://arxiv.org/abs/2007.09055
3. Shengpu Tang, Jenna Wiens. Model Selection for Offline Reinforcement Learning: Practical Considerations for Healthcare Settings. MLHC 2021. https://arxiv.org/abs/2107.11003
4. Justin Fu et al. Benchmarks for Deep Off-Policy Evaluation. ICLR 2021. https://arxiv.org/abs/2103.16596


**Summary Of The Paper:**

This paper examines the role of Bellman error as an objective function in offline reinforcement learning. The Bellman error measures how “closely” the estimated value function $Q$ satisfies the Bellman equation, while the value error measures how close the estimated value function $Q$ is to the true value function $Q^\pi$. In offline RL and OPE, we’re interested in value error, but since the true value function is unknown, common practice is to look at Bellman error.

The paper provides both theoretical analysis, and empirical experiments using two algorithms related to Bellman error (BRM and FQE). The main takeaways are: (i) zero Bellman error implies zero value error, but (ii) low Bellman error doesn’t imply low value error, making it a poor objective to optimize and a poor metric to compare using, (iii) the success of practical iterative value-based methods like FQE that optimizes Bellman error-like losses relies on generalization during training to reliably predict values for unseen state-action pairs.


**Summary Of The Review:**

This paper brings renewed attention to the role of Bellman error in offline RL, beyond the more widely known double sampling issue. In my opinion, Bellman error (or relatedly, TD error) is commonly thought of as the equivalent of *loss function* in supervised learning but for offline RL, and such analysis is much needed for the community and a good step at understanding the empirical success of deep RL and its fundamental difference compared to supervised learning. I would be strongly in favor of acceptance if the authors can aptly discuss how this contribution relates to the past work provided above.

---

> ### Author Response · Authors · 2021-11-17
> **Response to Reviewer oTrW**
>
> Thank you for your review and providing relevant references that we had overlooked. We have included the references (as well as a discussion of their differences to our work) and changed some of the writing in 4.2 to highlight how our work can be applied to the model selection setting.
>
> **Realizability/completeness**: Realizability and completeness are properties of the function class which are independent of the algorithm used to train them. For example, our function class of deep neural networks is (approximately) realizable as it is able to represent $Q^\pi$ (Fig 2). However, these properties are not necessarily reflected in generalization, as $Q^\pi$ being a member of the function class does not guarantee that it will be found or that generalization will be meaningful. When the function class is not realizable our main observations still hold (the target can still be closer to $Q^\pi$ even if it cannot capture it exactly) and are relevant (we are still relying on the target being accurate regardless of whether it is possible). While certainly relevant, we believe this analysis is mainly tangential to our work as our emphasis is on the significance of the objective itself, rather than convergence guarantees, sample complexity, or performance bounds.
>
> **Model selection**: Thank you for the references! The original submission does make observations on model selection, but we have added a few additional statements that may help connect our experiments to more practical observations. For example, we re-titled the paragraph “Is Bellman error a good proxy for value error?” in section 4.2 to “On-policy empirical Bellman error is insufficient to rank value functions” and adjusted some of the writing. We also have added a paragraph to the related work which discusses model selection and the references you have provided.
>
> **Model selection.differences**: One very key difference in our observations is that Fig 2 shows that Bellman error is a poor metric even when used with *on-policy* data, rather than with an off-policy/offline dataset as in the provided references, which is a stronger claim. A second difference is that these observations in the provided references are afterthoughts and only mentioned in the appendix. They are tested as baselines, shown that they are weakly related to policy performance, and then otherwise ignored. Our work greatly expands on understanding this phenomenon, providing theoretical insights and more exhaustive empirical analysis. We agree this is a great addition to the paper, also because model selection highlights the work of [1] which is older work which actually uses Bellman error for model selection. This is a nice example of where there exists a gap between theoretical and practical observations and highlights why our findings provide value to the community.
>
> [1] Farahmand, Amir-massoud, and Csaba Szepesvári. "Model selection in reinforcement learning." 2011.

---

> > ### Comment · Reviewer_oTrW · 2021-11-19
> > **Reply to author response**
> >
> > Dear authors,
> >
> > Thank you for responding to my questions and to others'. I appreciate the added discussion of how this contribution relates to offline RL, OPE, and model selection. I am willing to increase my rating to Accept. I have a few more thoughts/comments that I hope you can consider for the final version.
> >
> > - Construction of Theorem 1 (P4) and its proof (P16) looks somewhat similar to the performance difference lemma originally in Kakade & Langford 2002 (e.g. Proposition 2 of [this](https://nanjiang.cs.illinois.edu/files/cs598/note1.pdf)). Might be worth adding a reference to it.
> > - P5 below Eqn (8): I like the examples you gave here, but there's a disconnect between the toy example and empirical experiments that's worth pointing out. In the tabular setting, missing transitions may come off as easily avoidable; whereas in the function approximation setting, we are assuming that our function will generalize well to the entire $S \times A$, where in fact it may generalize poorly to relevant transitions not in the dataset. Without spelling this out, I feel like the readers might not appreciate the illustrative examples and how that connects to the practical situation.
> >
> > Suggestions on certain claims that might be too strong:
> > - Abstract: "This _eliminates any_ guarantees relating Bellman error to the accuracy of the value function." I would qualify this statement a bit more, right now it sounds like you're saying Bellman error and value error are not related in any way under all scenarios, which is probably not true.
> > - P9 Conclusion: "For a given finite dataset, we show there ~~exists~~ **can exist** infinitely many suboptimal value functions which satisfy the Bellman equation."
> > - P9 Related Work: I appreciate the added reference to Farahmand & Szepesvari 2011. Their paper posited minimizing Bellman error as the model selection objective without much justification other than prior work, considering the following except:
> > > Following previous works ...instead of directly working with policies, we consider the problem of ﬁnding an action-value function with a small Bellman error, which is supposed to facilitate the search for a good policy: When the Bellman error of an action-value function is zero (or very small) an optimal (respectively, good) policy can be obtained from the action value function with minimal effort.
> >
> >     Therefore, I wouldn't say your paper "contradicts" their result (the theory on minimizing Bellman objective still holds), but rather it cautions researchers to think carefully about whether Bellman error is the right metric (which your results show that it's not).

---

> > > ### Author Response · Authors · 2021-11-21
> > > **Thank you!**
> > >
> > > Thank you for the additional constructive comments and for adjusting your score! We’ve made some initial edits to the paper to address your comments and will consider your comments further for any final versions.

---

### Official Review · Reviewer_9cny · 2021-11-02

**Correctness:** 4
**Technical Novelty And Significance:** 3
**Empirical Novelty And Significance:** 3
**Recommendation:** 6
**Confidence:** 3

**Main Review:**

Strengths
---
The paper is of relatively high quality. Experiments are carried out with a clear purpose and the methodology seems sound.

The paper is well-written and well-structured. Theoretical results are presented cleanly and with useful exposition. I did not carefully check through the proofs in the appendix. Experimental results are also presented well and generally easy to follow.

And, as mentioned previously, these findings may be useful for a more complete understanding of when and why RL algorithms break down, especially in paradigms such as offline RL.


Weaknesses
---
I am not an expert in this area, so I am unsure as to whether all of the results (the theoretical results in particular) are fully novel or if prior work has discovered similar or roughly the same results. Additional discussion of this in the related works section would be appreciated. In particular, it seems like results showing that Bellman error can be optimized without optimizing the true objective have probably appeared in prior work, though again, I am not sure.

A potentially relevant line of recent work may be Q-learning and dynamic programming approaches to offline RL. [1] may serve as a useful survey here.

Some greater analysis of the experiments would also be generally helpful. In Fig 2, are there potentially some conclusions missing due to incomplete training, e.g., some of the learning curves still exhibit a downward trend at 1 million steps? And explanations for some of the outliers, e.g., FQE HalfCheetah value error and BRM Hopper Bellman error, could prove insightful. A similar comment holds for Fig 3.

More vaguely, the tables and Fig 4 are harder to follow. Adding just a couple of sentences in the main text to more explicitly describe the tables and Fig 4 would likely fix this.

Even more vaguely, but perhaps also most importantly, I still have lingering concerns regarding the significance of this work. Some examples of how this work may be significant are: the work provides insight for how to devise a better RL algorithm, or the work sheds light on why some prior RL algorithms work better than others. Currently in the text, I do not see clear arguments for either of these or other examples of what may be the immediate impact of this work. Should the authors have additional thoughts here regarding how the work may influence additional work (or provide insight into prior work), that would be appreciated.


[1] Levine et al, "Offline Reinforcement Learning: Tutorial, Review, and Perspectives on Open Problems." arXiv 2005.01643.

**Summary Of The Paper:**

This paper studies the Bellman equation commonly used in reinforcement learning (RL) algorithms. The typical motivation behind using the Bellman equation to design RL objectives is that uniformly driving the Bellman error to 0 implies that the true value function has been learned, i.e., the value error is 0. However, as this paper demonstrates both theoretically and empirically, a number of issues may arise before this point in practical regimes. For example, Bellman error and value error may not be well correlated, as shown via experiments evaluating minimizing the Bellman error directly versus fitted Q-evaluation. Furthermore, when evaluated using off-policy data, Bellman error may not even be well correlated with value error for different runs of the same algorithm. These findings may be useful for a more complete understanding of when and why RL algorithms break down, especially in paradigms such as offline RL.

**Summary Of The Review:**

Primarily due to concerns about significance, I am initially recommending a weak accept of this paper. I am happy to engage in discussion with the authors and other reviewers in order to reach a more confident final recommendation.

---

> ### Author Response · Authors · 2021-11-17
> **Response to Reviewer 9cny**
>
> Thank you for your review. We have made a number of adjustments to the paper to address your concerns, and hope this improves the paper.
>
> **Related work**: The second paragraph of our related work section highlights the work that we believe to be most closely related. While we believe that experienced readers may have some sense of familiarity with the main results (reviewer ZRun highlighted this) we do not believe there is any existing work which makes our claims. There has been work on minimizing alternate modified Bellman objectives, but this is largely centered around tackling the double sampling issue. We have added several references to the related work.
>
> There is also the common belief that minimizing value error would consequently result in minimizing the Bellman error. This comment gave us an idea for an additional experiment, where we added a Monte Carlo estimator to Fig 2 (policy evaluation with on-policy data) and show that while this (unsurprisingly) achieves a low value error, it also has higher Bellman error than the other method. We believe this helps highlight the main point we were trying to make with Fig 2 (that overfitting to Bellman error is possible, even with on-policy data). Thank you!
>
> **Offline RL**: This is definitely relevant. The original submission had some related work on offline RL already, but we’ve expanded the paragraph and included the mentioned reference.
>
> **Comments on Fig 2 & 3**: We’ve extended the curves in Fig 2 in Appendix D.4 to 10 million time steps. That being said, the goal of Fig 2 is not to show that one method is necessarily better than another, rather that it is possible to have functions, evaluated with on-policy data, where the relative ordering of value error and Bellman error is not consistent. Since any function $\mathcal{S} \times \mathcal{A} \rightarrow \mathbb{R}$ is a valid value function this observation is true regardless of whether training is complete. This is why the existence of outliers (in Fig 2) does not necessarily affect our analysis (as we are not arguing that there is an inverse relationship between Bellman error and value error, simply that it is possible in “realistic” situations). In Fig 3, the outliers occur due to the value estimate of FQE diverging, so we’ve added a comment to the table caption to explain that.
>
> **Descriptions of Fig 4 and Tables**: Thanks for pointing this out, we’ve expanded the intext description for each of these.
>
> **Significance**: We believe our work is significant in several ways! We have refactored the discussion in Section 4.2 to highlight the significance of our work. Some places our observations are relevant:
> - Algorithm design: We show that the Bellman equation is a poor off-policy objective. This means that algorithms which attempt to minimize Bellman error (or some modified variant) are unlikely to work with purely off-policy data.
> - Model selection: We show that Bellman error is insufficient to rank value functions. This means given a set of value functions, we cannot determine the best one via the Bellman error *even when using on-policy data*. This is a contradiction of more traditional results [1] (and corroborated by some of the references provided by Reviewer oTrW).
> - Performance: We provide insight into the performance gap between BRM and FQE methods and explain why FQE is a favorable approach.
> - General interest: The Bellman equation is a fundamental concept in RL & widely used by the community. We believe that any analysis that improves our understanding of the Bellman equation provides value to the community, even without immediate impact.
>
> [1] Farahmand, Amir-massoud, and Csaba Szepesvári. "Model selection in reinforcement learning." 2011.

---

### Official Review · Reviewer_ZRun · 2021-11-08

**Correctness:** 3
**Technical Novelty And Significance:** 2
**Empirical Novelty And Significance:** 2
**Recommendation:** 3
**Confidence:** 5

**Main Review:**

## Strong Points
Significance: I agree with the authors that the empirical demonstration of comparing Bellman error (training loss) vs value error (testing loss) is less considered in off-policy evaluation community and should receive higher emphasis. This paper may bring potential attention to the community to rethink of how to view Bellman error into the design of the algorithm.

Counterexample Construction: The paper construct several simple MDP as counterexample that the Bellman error is not with aligned with the value error.

Empirical evaluation: The empirical evaluation of this paper is extensive and provides useful insight. Reader can save their time to try BRM on TD3 (which is extensively studied in this paper and is failed)

## Weak Points
Novelty: The direct use of BRM fails to solve the off-policy evaluation has been widely known in the community. The main conclusion in this paper tries to emphasis on the fail of generalization of the Bellman error because of the finite dataset. However, I believe there are a lot of off-policy evaluation papers, especially those with high confidence interval estimation papers are trying to take the finite sample into account. Without any model assumption on either the value function, dynamics, rewards or density function, no algorithm can learn the region we have not observed and the value error can be arbitrary (like what you said in the example in Fig. 1) and therefore we need to make reasonable assumption in order to generalize the existing region's information. If we can assume the value function to be in a linear class, or smooth or with a known Lipschitz constant we can definitely be able to generalize (check theoretical papers for off-policy evaluation).

Quality: I like the motivation of the paper, but when I go deep into the main body of the paper, I do not find too many new things to me and feel disappointed. I would expect a new algorithm can be provided to alleviate the difference between Bellman error and the value error by either under reasonable assumption or a new way of using the Bellman information. Another thing I would expect the explanation of the comparable success of FQE. Pro. 3 looks weak to me and only said that if the sign is the same, we can expect improvement. however, since the target network $Q_{\bar \theta}$ is also randomly initialized the same as $Q_\theta$, we don't know much about the sign. Another point is that in real practice the target network can also updated in each iteration with a slower rate, so we cannot think this is similar to a fixed regression version in Eq. (13).

**Summary Of The Paper:**

The paper tries to argue that the Bellman error cannot be trusted as a good metric or objective for off-policy evaluation. To support the argument, the paper provides theoretical analysis as well as extensive empirical analysis. Theoretical analysis gives a few example on the relation between the true value error and the Bellman error, which could be arbitrary especially under finite sample cases. Empirical results show that even when the Bellman residual is minimized well, the value error could be large in various of settings, in comparison, FQE consistently outperform BRM even when the Bellman error is much larger.

**Summary Of The Review:**

In sum, I think this is an interesting paper and the topic worth further investigating. But so far the paper is not ready enough for the bar of the top conference like ICLR. I would vote for rejection at this moment and encourage the authors to further improve on the idea and submit to another venue.

---

> ### Author Response · Authors · 2021-11-17
> **Response to Reviewer ZRun**
>
> Thank you for your comments, while we politely disagree with some of your key points, we do appreciate the detailed review and have made several writing adjustments within the paper to reflect your concerns.
>
> **Novelty**: Our belief is that the use of BRM is not popular in the OPE community due to the double sampling problem, which we avoid by considering deterministic environments. This paper is not about disqualifying BRM as an OPE method, and rather is about analyzing the Bellman equation as a metric/objective/proxy for value accuracy when evaluated with finite and off-policy data. Note that our observations extend beyond off-policy BRM. In Fig 2 we show that the *on-policy* Bellman error is insufficient to rank value functions by value error.
>
> **Novelty.References**: It’s unclear if the reviewer is also suggesting that our findings regarding the Bellman equation are not novel. Were there any specific references that we overlooked? While we are not surprised that an experienced reader would have a sense of familiarity with our findings, to the best of our knowledge, there are no papers which study the Bellman equation evaluated with finite/off-policy data, and present similar conclusions. Given the importance and widespread use of the Bellman equation, particularly with off-policy data (in the DRL community: such as DQN/DDPG and their successors, as well as the offline RL community), providing concrete evidence to a phenomenon which is widely known but not studied, can have significant value. As the 3 other reviewers did not mention novelty concerns, we suspect these findings are not necessarily widely known to the RL community at broad. Please correct us if we are mistaken.
>
> **Novelty.Finite data**: While there are other OPE papers which consider finite data via confidence interval estimation, these are typically (1) based on IS methods, rather than the Bellman equation, (2) are focused on developing algorithmic improvements rather than studying the base algorithm. Since the Bellman equation is commonly used in finite data, and off-policy settings, we believe that understanding the vanilla Bellman equation, rather than some improvement/adjustment, provides value to the community. Again, if there are any specific references, we would happily include them in the paper. One possible miscommunication is a sentence in the introduction which we have modified as follows:
> “Our findings point to an underappreciation of the importance of finite data” -> “Our findings point to an underappreciation of the importance of finite data in widely used objectives”
>
> **Quality**: One objective of the paper is to explain the performance gap between BRM and FQE. This is tackled twofold, firstly we explain why BRM performs poorly (a meaningless off-policy objective and early convergence) (end of 4.2) and secondly discuss that the iterative objective in FQE is more strongly related to value error & that FQE allows for generalization in the target, while BRM inhibits it (4.3). We feel as if this is a communication issue and have adjusted some of the writing (final paragraph of 4.2 and 4.3) to communicate these ideas more clearly.

---

### Author Response · Authors · 2021-11-29
**Final comments before the discussion period ends**

Thank you again for the reviewers & AC for their insightful comments & time. Since two reviewers have yet to engage in discussion, we would like to highlight how we modified the paper to address their feedback (as well as the feedback of the other two reviewers).

 - Added discussion to related work on minimizing Bellman error (**9cny**, **ZRun**), offline RL (**9cny**), and model selection (**oTrW**), highlighting their relevance and/or differences.
 - Rewrote the presentation of results in section 4.2 to highlight the significance of our contributions more clearly (**9cny**, **ZRun**).
 - Added MC value estimation to Figure 2 to better emphasize how our results also impact on-policy learning, rather than just off-policy learning (**9cny**, **ZRun**).
 - Expanded Figure 2 to 10 million time steps in the Appendix (**9cny**).
 - Added explanations for outliers in Fig 2 and 3 and expanded the in-body description of the tables and Figure 4 in the main body of the paper (**9cny**).
 - Adjusted the wording on claims which came across too strongly (**ZRun**, **oTrW**).
 - Highlighted the role of model selection in Section 4.2 (**oTrW**).
 - Added descriptions of how our theoretical results affect other MDP settings (**fSvY**).

We understand that many reviewers (especially ones who are also authors) have limited time. Regardless, even without their confirmation, we feel that these adjustments/additions both improve the paper and address the concerns of the reviewers.

We are more than open to further improving the paper if any additional comments arise after the discussion period has ended (or within the remaining time) and hope our current efforts at improving the paper provides good faith that we are committed to doing so.

---

### Decision · Program_Chairs · 2022-01-20

**Decision:**

Reject

**Comment:**

This paper studies whether the Bellman error is a good metric to reflect the quality of value function estimation, focusing on finite-sample off-policy data sets. Both theoretical analyses and empirical experiments have been provided, showing that the Bellman error is often not the right metric to consider. However, while I appreciate the authors' theoretical attempts, the current theoretical contributions are not deep/significant enough. As the reviewers mentioned, the failure of the direct use of BRM is not surprising given the insufficiency of data (namely, no algorithm can make predictions on completely unseen regions unless further modeling structure is present). The authors might want to further strengthen their theory along this important direction.